# Single-droplet surface-enhanced Raman scattering decodes the molecular determinants of liquid-liquid phase separation

Anamika Avni[1,2,4], Ashish Joshi[1,3,4], Anuja Walimbe[1,3], Swastik G. Pattanashetty[1,3] & Samrat Mukhopadhyay [1,2,3] ✉

Biomolecular condensates formed via liquid-liquid phase separation (LLPS) are involved in a myriad of critical cellular functions and debilitating neuro-degenerative diseases. Elucidating the role of intrinsic disorder and con-formational heterogeneity of intrinsically disordered proteins/regions (IDPs/IDRs) in these phase-separated membrane-less organelles is crucial to under-standing the mechanism of formation and regulation of biomolecular con-densates. Here we introduce a unique single-droplet surface-enhanced Raman scattering (SERS) methodology that utilizes surface-engineered, plasmonic, metal nanoparticles to unveil the inner workings of mesoscopic liquid droplets of Fused in Sarcoma (FUS) in the absence and presence of RNA. These highly sensitive measurements offer unprecedented sensitivity to capture the crucial interactions, conformational heterogeneity, and structural distributions within the condensed phase in a droplet-by-droplet manner. Such an ultra-sensitive single-droplet vibrational methodology can serve as a potent tool to decipher the key molecular drivers of biological phase transitions of a wide range of biomolecular condensates involved in physiology and disease.

Biomolecular condensation via liquid-liquid phase separation (LLPS) offers an exquisite mechanism for spatiotemporally-controlled organization and compartmentalization of cellular constituents into highly dynamic, permeable, liquid-like, tunable, mesoscopic, non-stoichiometric supramolecular assemblies known as membrane-less organelles[1–10]. These on-demand non-canonical organelles containing proteins and nucleic acids are present both in the cytoplasm and nucleus and include nucleoli, stress granules, P granules, nuclear speckles, and so forth. A growing body of exciting research suggests that highly flexible intrinsically disordered proteins/regions (IDPs/IDRs) containing low-complexity regions and prion-like domains that

offer conformational heterogeneity, distributions, and multivalency are excellent candidates for intracellular phase separation. A unique combination of these sequence-dependent features governs the making and breaking of promiscuous and ephemeral intermolecular interactions such as electrostatic, hydrophobic, hydrogen bonding, dipole-dipole, π–π, and cation−π interactions resulting in liquid-like behavior[3,6,11–17]. While these highly dynamic condensates are proposed to be involved in a range of critical cellular functions, their transitions into less dynamic gel-like or solid-like aggregates containing more persistent interchain interactions are linked to debilitating neurode-generative diseases. Therefore, it is imperative to decipher the

[1]Centre for Protein Science, Design and Engineering, Indian Institute of Science Education and Research (IISER) Mohali, Punjab, India. [2]Department of Chemical Sciences, Indian Institute of Science Education and Research (IISER) Mohali, Punjab, India. [3]Department of Biological Sciences, Indian Institute of Science Education and Research (IISER) Mohali, Punjab, India. [4]These authors contributed equally: Anamika Avni, Ashish Joshi. ✉e-mail: mukhopadhyay@iisermohali.ac.in

fundamental molecular interactions of phase transitions involved in functions and disease[18]. A multitude of spectroscopic and microscopic methodologies have been employed to unveil the key biophysical principles of phase separation resulting in the formation of liquid droplets. For instance, high-resolution microscopic tools such as confocal, super-resolution, and high-speed atomic force microscopy can directly probe the properties within individual liquid droplets[19,20]. However, these tools do not allow us to access the wealth of molecular information in a residue-specific manner. In contrast, the high-resolution structural methods such as nuclear magnetic resonance (NMR) and small-angle X-ray scattering (SAXS) provide the atomic-resolution details of the condensed phase architecture[21–23]. However, these ensemble structural methods are not capable of yielding molecular insights from the condensed phase of individual droplets. Therefore, a method that combines these capabilities to capture residue-specific structural information at a single-droplet resolution is essential to characterize and quantify the key molecular determinants in a droplet-by-droplet manner.

Vibrational Raman spectroscopy performed in a microscopy format allows us to uniquely and elegantly combine the aforesaid capabilities to obtain the protein structural information from a well-defined spatial location by focusing the laser beam into a sub-micron spot. Such non-invasive and label-free laser micro-Raman measurements permit us to access the wealth of structural information by monitoring a range of bond vibrational frequencies while retaining the spatial resolution[24–26]. However, owing to a low Raman scattering cross-section, Raman spectroscopy is a highly insensitive technique, especially for biomolecules under physiological conditions in aqueous solutions[27]. Additionally, the high laser power required for Raman spectroscopic detection can be detrimental to soft biological samples. The low-sensitivity issue in Raman scattering can be circumvented by a near-field plasmonic enhancement by metallic nanostructured substrates giving rise to high electromagnetic/chemical enhancement of Raman signals even at extremely low analyte concentrations. This surface-sensitive technique known as surface-enhanced Raman scattering (SERS) that can provide several orders of magnitude increase in the Raman scattering cross-section allowing single-molecule detection and characterization even at a much lower laser power[28–32]. In the present work, we have developed an ultra-sensitive single-droplet SERS methodology that can illuminate the unique molecular details of the polypeptide chains within individual phase-separated protein liquid droplets. For our studies, we have used Fused in Sarcoma (FUS), which is one of the most intensely studied RNA-binding proteins containing archetypal prion-like low-complexity domains and hence one of the best prototypes of phase-separating proteins. The human genome encodes approximately 30 FUS-family proteins that are known to be involved in critical functions such as mRNA splicing, DNA damage repair, formation of stress granules as well as in deadly neurodegenerative diseases such as ALS (amyotrophic lateral sclerosis) and FTD (frontotemporal dementia)[33–38]. Here we show that upon liquid phase condensation, surface-coated, SERS-active, nanosphere substrates get spontaneously encapsulated within the protein-rich condensed phase and generate plasmonic hotspots that permit us to capture the inscrutable workings of FUS condensates with unprecedented sensitivity in the absence and the presence of RNA.

## Results

### Experimental design for single-droplet vibrational Raman spectroscopy

Our laser micro-Raman system consists of several components, namely, an excitation source comprising of a near-infrared (NIR) laser, an integrated microscope spectrometer consisting of a combined system of lenses, mirrors, filters, and a diffraction grating, and a charge-coupled device (CCD) detector (Fig. 1). This integrated optical setup allows us to irradiate the sample and filter out the (elastic) Rayleigh scattered light and further collimate the (inelastic) Raman scattered light onto the detector to obtain a Raman spectrum. Such a design permits us to focus the laser beam of a suitable power using an objective lens into a small sub-micron spot-size within a single protein-rich droplet and acquire (regular) normal single-droplet Raman spectra. For ultra-sensitive SERS measurements, we observed that surface-modified metal nanoparticles get spontaneously encapsulated into liquid droplets, as evident by an independent confocal fluorescence imaging experiment (Fig. 1). The single-droplet SERS methodology allows us to obtain highly enhanced Raman signals within individual liquid droplets of FUS.

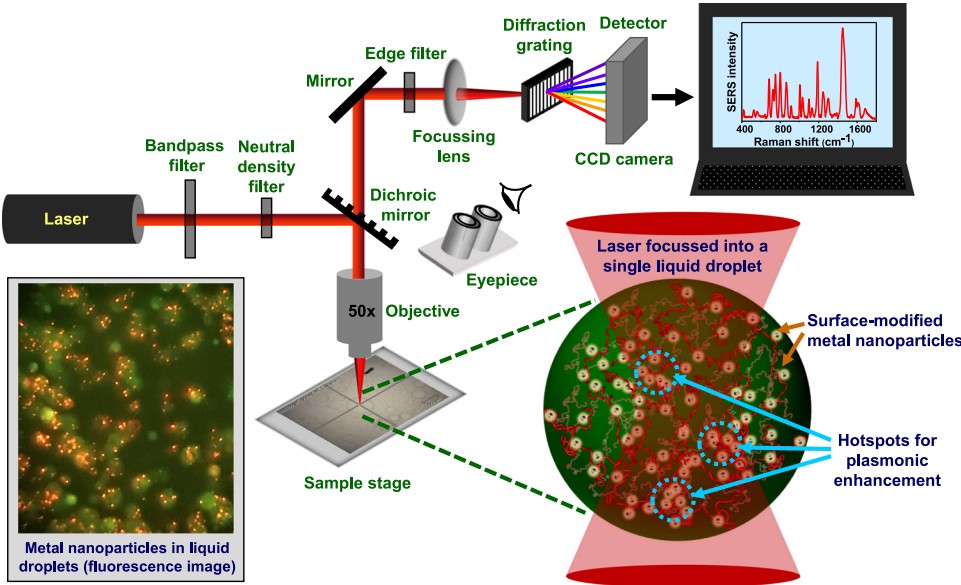

**Fig. 1 | A sketch of the optical setup and a schematic of single-droplet normal and surface-enhanced Raman spectroscopy (SERS).** A near-infrared (NIR) laser gets focussed within each protein-rich FUS droplet encapsulating surface-functionalized silver nanoparticles through an integrated system of lenses, mirrors, and filters. Hotspots are generated within the droplets causing optical enhancement of Raman signals detected by a CCD detector. A fluorescence image taken from the eyepiece using a camera is also included to show the encapsulation of nanoparticles in the condensates.

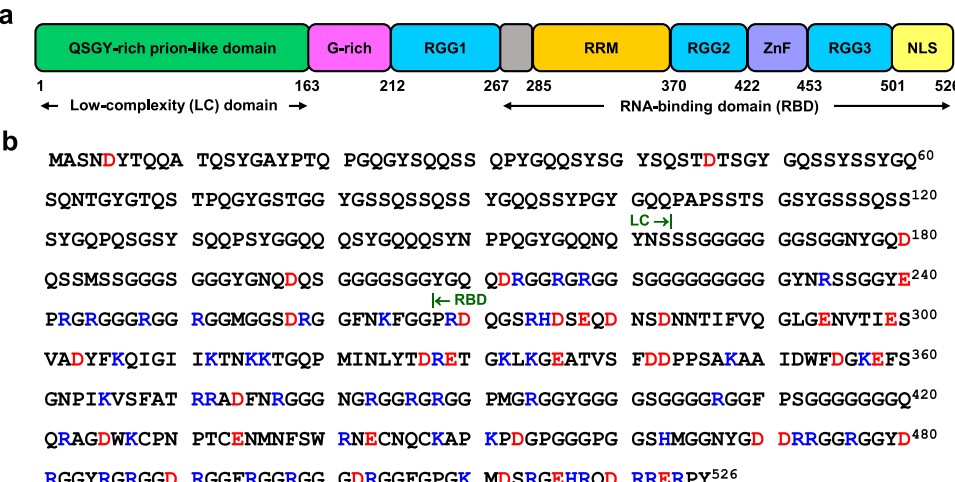

**Fig. 2 | Domain architecture and amino acid sequence of full-length FUS.**
**a** Schematic representation of full-length FUS showing all the segments and domains. **b** Amino acid sequence of full-length FUS. The positively and negatively charged residues are highlighted in blue and red, respectively.

## Single-droplet normal Raman spectroscopy of FUS condensates

FUS consists of an intrinsically disordered N-terminal low-complexity (LC) prion-like domain (residues 1–163) and a C-terminal RNA-binding domain (RBD) containing both disordered and α-helical secondary structural elements (residues 267–526) (Fig. 2). The RBD contains two RGG-rich stretches, an RNA-recognition motif (RRM), and a zinc finger domain and carries a net-positive charge (+10.8) at physiological pH. The intrinsically disordered LC domain is the primary driver of phase separation, and the presence of the RBD greatly enhances the propensity for phase transitions under physiological conditions[14,34–36]. We recombinantly expressed FUS with a cleavable N-terminal maltose-binding protein (MBP) tag as described earlier[37]. We then induced LLPS by cleaving the MBP tag using TEV protease, which resulted in the condensation of a homogeneously mixed aqueous solution to liquid droplets as reported previously[37]. The condensed droplet phase was devoid of cleaved MBP as observed before (Supplementary Fig. 1)[37]. Next, we set out to perform single-droplet vibrational Raman experiments on these FUS liquid droplets by focusing the laser beam within individual liquid droplets, one at a time. Prior to performing more advanced and involved SERS experiments, we carried out (regular) normal single-droplet Raman spectroscopy by focusing a high-power (500 mW) 785 nm laser beam using a ×100 objective. These studies allowed us to characterize the condensed phase by recording the Raman scattering bands for different vibrational modes of the polypeptide chains in a droplet-by-droplet manner (Fig. 3a). Single-droplet Raman spectra were dominated by amide I (1620–1700 cm⁻¹), amide III (1230–1300 cm⁻¹) as well as bands due to different vibrational modes of aromatic amino acids (phenylalanine, tyrosine, and tryptophan) in addition to several other aliphatic sidechain vibrations (Fig. 3b, Supplementary Table 1, 3)[27,39–41]. Amide I (1620–1700 cm⁻¹) arise primarily due to C=O stretching vibrations and amide III (1230–1300 cm⁻¹) represents C–N stretching and N–H bending vibrations of the polypeptide backbone. These amide bands in a protein Raman spectrum are the typical marker bands for secondary structural elements[42]. The condensed phase showed a broad amide I band centered at ~1671 cm⁻¹ with a full-width at half-maximum (FWHM) ~59 cm⁻¹ representing a considerable conformational distribution. This was further supported by the amide III band at ~1262 cm⁻¹ representing highly disordered conformers within the condensed phase. Further, to decode the sidechain environment, we inspected the intensity ratio at 850 cm⁻¹ and 830 cm⁻¹ ($I_{850}/I_{830}$) of the tyrosine Fermi doublet that is observed due to Fermi resonance between the ring breathing vibration and overtone of an out-of-plane ring-bending vibration of the phenolic ring of tyrosine. Therefore, this ratio is an indicator of solvent-mediated

hydrogen bonding propensity of the phenolic (−OH) group and is a measure of the water accessibility of tyrosine residues[43]. The $I_{850}/I_{830}$ ratio is typically ≥2 for a well-solvated tyrosine and this ratio was found to be ~0.5 for FUS droplets indicating considerable solvent protection possibly due to the participation of tyrosine residues in π–π stacking and/or cation–π interactions in the dense phase[44]. Another important sidechain band is the tryptophan Raman band typically observed at ~880 cm⁻¹ that arises due to the indole N–H bending and is often used to probe the environment and is a measure of the hydrogen bonding strength between the N–H of the indole ring with the surrounding solvent molecules. This band is highly blue-shifted to 891 cm⁻¹ in droplets indicating a reduced hydrogen bonding propensity of the N–H group with the surrounding water molecules implying an apolar microenvironment in the vicinity of tryptophan residues[45]. Additionally, we observed a tryptophan band at 767 cm⁻¹ that corresponds to the indole ring breathing and is used as a marker for cation–π/CH–π interactions[46,47]. Therefore, these observations might potentially indicate the presence of π–π/cation–π interactions within FUS condensates. Additionally, we observe considerable variation in the intensity of Raman bands in our single-droplet measurements (Supplementary Fig. 2). Since these protein droplets possess a wide range of sizes having heterogeneous protein conformations as well as varying extent of protein-protein interactions, we do expect to observe this intensity variation due to a high degree of conformational heterogeneity within the dense phase of droplets. We also observed a broad band at around 540 cm⁻¹ corresponding to characteristic backbone deformations due to the presence of a large number of highly flexible glycine residues in FUS[48,49]. Taken together, this set of normal single-droplet Raman experiments indicate conformational heterogeneity and intrinsic disorder of FUS within the droplets. These results also revealed the involvement of aromatic side chains of tyrosine and tryptophan residues in the chain collapse and condensation of FUS corroborating previous findings[23]. Next, in order to enhance the sensitivity as well as to detect and characterize weaker and hidden vibrational signatures in FUS and FUS-RNA condensates, we set out to perform single-droplet SERS measurements.

## Preparation of surface-modified nanoparticles for single-droplet SERS

The predicted net charge of FUS is +14.5 at physiological pH and we thus postulated that it could electrostatically interact with surface-modified negatively charged metal nanoparticles offering us an excellent system to study SERS within biomolecular condensates. To test our hypothesis, we started with the preparation of a suitable plasmonic

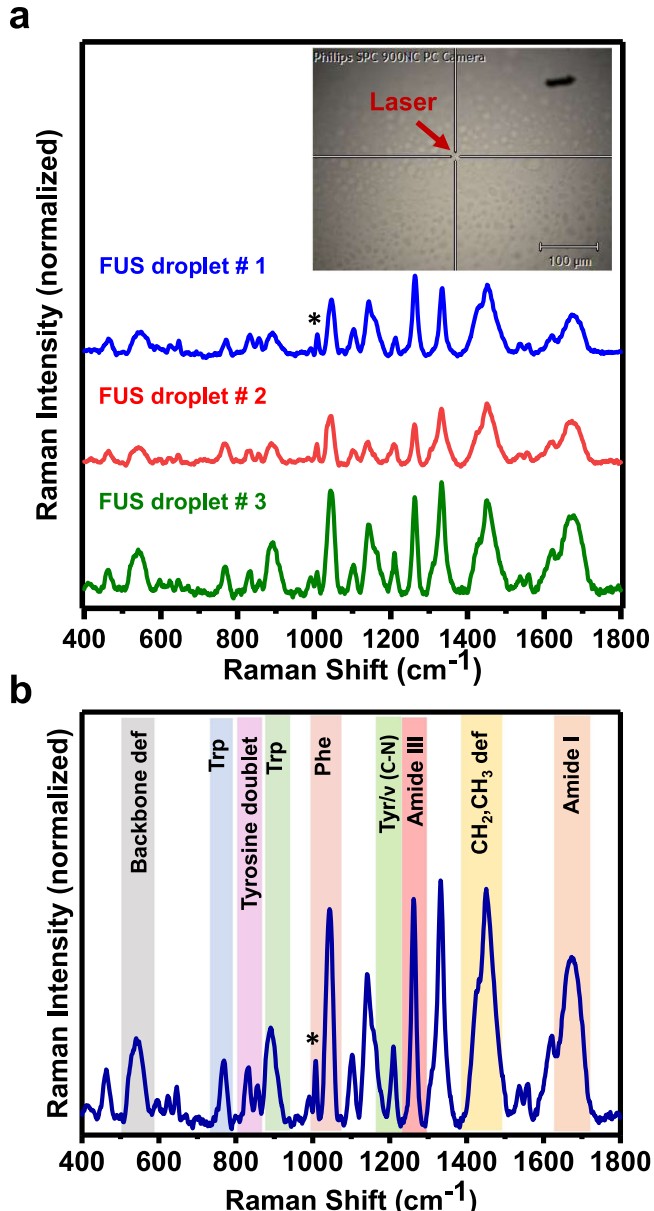

**Fig. 3 | Single-droplet normal Raman spectroscopy. a** Representative single-droplet normal Raman spectra of a few individual FUS droplets (spectra recorded at 500 mW laser power, ×100 objective; number of droplets, $n = 3$). Inset shows single droplets of FUS focussed through the Raman microscope. Arrowhead shows the focal point of the NIR laser within the droplet. **b** Average Raman spectrum of the FUS condensed phase. All spectra are normalized with respect to the phenylalanine ring breathing band at 1007 cm⁻¹ marked by an asterisk. See Methods for details of data acquisition, processing, and analysis.

**SERS substrate.** We chose silver nanoparticles (Ag NPs) for our experiments due to their high SERS activity and greater electromagnetic enhancements compared to other plasmonic nanomaterials. We prepared Ag NPs by a standard protocol of reduction of $AgNO_3$ by sodium citrate and characterized them using UV–visible absorption spectroscopy, transmission electron microscopy (TEM), and zeta ($\zeta$) potential measurements (Supplementary Fig. 3)[50]. The absorption spectrum showed a single absorption band at ~415 nm which corresponds to spherical nanoparticles with a diameter of ~30 nm (Supplementary Fig. 3a)[51]. We next functionalized these Ag NPs with iodide to form iodide-modified silver nanoparticles (Ag IMNPs). This halide modification of nanoparticles was performed to get rid of the overwhelming citrate peaks in the Raman spectrum and aid in better attachment of the

polypeptide chains to the negatively charged silver nanospheres (Supplementary Fig. 3b)[50]. Ag IMNPs exhibited a single absorption band with $\lambda_{max}$ at ~418 nm indicating a similar diameter of surface-modified nanoparticles (~30–40 nm). The size was then confirmed by TEM which revealed nanospheres with an average diameter of ~30 nm (Supplementary Fig. 3c). We next carried out zeta potential measurements to determine the effective charge on the surface of nanoparticles. The zeta potential of Ag IMNPs was −22 mV indicating an overall negative surface charge that stabilizes the nanoparticles preventing them from agglomerating into large-sized colloids (Supplementary Fig. 3d). Further, Raman spectra of these surface-modified nanoparticles showed a single band at 110 cm⁻¹ that corresponds to Ag−I bond indicating monolayer coating of Ag NPs (Supplementary Fig. 3e)[50].

We next set out to investigate the interaction of FUS with Ag IMNPs and to check their colloidal stability in the presence of the protein. We recorded UV−vis absorption spectra of Ag IMNPs in the dispersed and condensed phase of FUS at 10 and 30 min time points and observed a small red-shift of the absorption maxima with a slight broadening of the band as compared to only Ag IMNPs in buffer (Supplementary Fig. 4a, b). We chose to record the absorption spectra at these two different time points since our next set of single-droplet SERS measurements required this time interval for data acquisition. The addition of FUS monomer to Ag IMNPs causes an increase in the absorbance along with a red-shift and broadening of the absorption peak. This suggests that the NPs bind electrostatically to the monomeric protein and form small nanoparticle clusters without altering the stability of nanoparticles[52]. In the droplet state, the broadening of the absorption peak with a drop in the absorbance indicates more clustering of NPs within the condensed phase generating hotspots necessary for plasmonic enhancements in Raman measurements. In order to test if the LLPS propensity and the structure of FUS remained unaltered in the presence of nanoparticles, we performed turbidity and CD experiments (Supplementary Fig. 4c, d). Additionally, our microscopy studies indicated that nanoparticles do not alter the phase separation propensity of FUS and that Ag IMNPs are completely encapsulated with FUS droplets (Supplementary Fig. 4e−g). Furthermore, we performed size distribution analyses that showed no significant changes in the droplet dimension in the absence and the presence of nanoparticles (Supplementary Fig. 4h). To visualize the presence of nanoparticles within the condensates, we performed two-color confocal fluorescence imaging that revealed the uptake of nanoparticles within the condensates (Fig. 4a, b and Movie S1). Fluorescence recovery after photobleaching (FRAP) experiments on fluorescently labeled FUS droplets revealed no significant difference in the rate of recovery in the absence and the presence of nanoparticles (Fig. 4c). These results indicated that the droplet interior remained mobile in the presence of nanoparticles and the overall material property of FUS condensates remained unaltered in the presence of the SERS substrate. Together, this set of experiments suggested that FUS electrostatically interacts with surface-coated silver nanoparticles that get preferentially encapsulated into the dense phase of liquid droplets while keeping the internal mobility nearly intact. Therefore, these silver nanoparticles can act as an ideal SERS substrate for Raman enhancements within biomolecular condensates. We next directed our efforts to perform ultra-sensitive SERS measurements within individual droplets.

**Single-droplet SERS within FUS condensates**
In order to record single-droplet SERS spectra, LLPS was set up in the presence of 100 pM Ag IMNPs, and a 785-nm NIR laser beam (5 mW) was focused into individual nanoparticle-containing liquid droplets using a ×50 objective (Supplementary Fig. 4g). We achieved an enhancement in the order of ≥10⁴ using amide III as a reference peak. We would like to note that this is an approximate (lower bound) estimate of the enhancement factor due to extremely weak signals from the droplets in the absence of nanoparticles using low power and a ×50

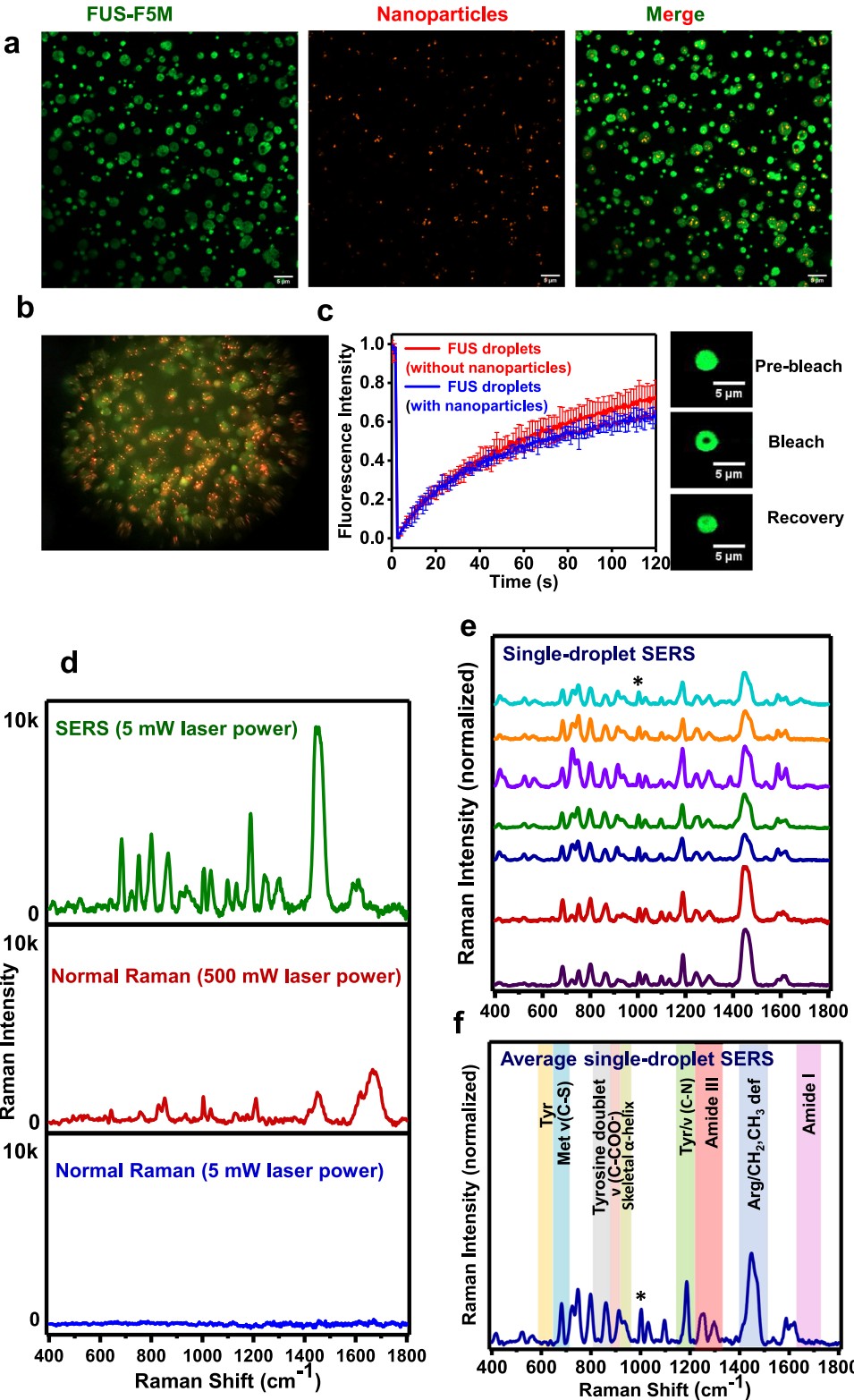

objective (Fig. 4d). The enhancement is much higher than $10^4$ for the peaks that are not visible in normal Raman. Interestingly, the amide I band in our SERS spectra was not enhanced, and therefore, was not visible in such a low power illumination (Fig. 4e, f). According to selection rules, signal enhancement in SERS depends on the orientation of the analyte on the surface of nanoparticles and varies inversely with the twelfth power of the distance of the target analyte from the nanoparticle surface[53]. We speculate that the interaction of bulky amino acid side chains of FUS with the negatively charged Ag IMNPs

might potentially orient the backbone C=O group away from the critical near-field required for the plasmonic enhancement[54]. Nevertheless, amide III was visible at 1246 cm⁻¹ and 1298 cm⁻¹ corresponding to nonregular/turn structures and α-helices, respectively (Fig. 4e, f). Although histidine tautomeric doublets can appear in this region, FUS being a 526-residue protein, contains 525 amide bonds and only three histidine residues. Therefore, the contribution from histidine tautomeric doublets can be much lower compared to amide vibrations. Several orders of magnitude signal enhancement allowed us to

**Fig. 4 | Spontaneous encapsulation of iodide-modified silver nanoparticles (Ag IMNPs) in FUS liquid droplets and single-droplet SERS. a** Confocal images depicting encapsulation of iodide-modified silver nanoparticles (Ag IMNPs) within fluorescein-5-maleimide labeled FUS droplets. See Movie 1 for 3D Z-stack images of droplets containing nanoparticles. The imaging was performed thrice with similar observations. **b** Image clicked through the eyepiece using a camera indicating the encapsulation of Ag IMNPs within FUS droplets (a similar image is also shown in a scheme in Fig. 1). **c** FRAP kinetics of multiple droplets (~1% Alexa488-labeled protein) in the absence (red) and presence of nanoparticles (blue). The data represent mean ± s.d. for $n = 3$ independent experiments. Fluorescence images of droplets during FRAP measurements are shown on the right. **d** Single-droplet SERS (5 mW

laser power), single-droplet normal Raman (500 mW laser power), and single-droplet normal Raman spectra (5 mW laser power) using ×50 objective lens. The observed enhancement was of the order of $\geq 10^4$ times using Amide III as a reference peak. **e,** Stacked single-droplet SERS spectra of 7 FUS droplets in the presence of Ag IMNPs (spectra recorded at 5 mW laser power with a ×50 objective). **f,** Average single-droplet SERS spectra from individual droplets encapsulating Ag IMNPs ($n = 7$). See "Methods" for experimental details and Supplementary Table 3 for all the band positions. All spectra are normalized with respect to the phenylalanine ring breathing band at 1003 cm$^{-1}$ marked by an asterisk. See Methods for details of data acquisition, processing, and analysis.

observe these structures and is possibly caused by electrostatic interactions between the negatively charged surface of Ag IMNPs and arginine-rich positively charged RBD containing these secondary structural elements. In addition, a significant enhancement was also observed for backbone C–C stretch of α-helices at 938 cm$^{-1}$. Surprisingly, closer inspection of the tyrosine Fermi doublet showed that the lower wavenumber band at 830 cm$^{-1}$ did not show significant enhancement while the higher wavenumber band showed enhancement and shifted to 863 cm$^{-1}$. This may be attributed to the SERS selection rules, according to which the polarizability component of the 830 cm$^{-1}$ mode may not be perpendicular to the metal surface. Further, single-droplet SERS spectra were dominated by bands at 683 cm$^{-1}$ [methionine/δ(CH)]; 724 cm$^{-1}$ (methionine); 749 cm$^{-1}$ (tryptophan); 915 cm$^{-1}$ ν(COO$^-$); 1003, 1032 cm$^{-1}$ (phenylalanine); 1246, 1298 cm$^{-1}$ (Amide III); 1588 cm$^{-1}$ (phenylalanine/tryptophan/histidine); 1621 cm$^{-1}$ (tyrosine) (Supplementary Table 3). Additionally, we observed a highly enhanced band at 1447 cm$^{-1}$ and a shoulder at 1465 cm$^{-1}$ that is assigned to the bending δ(NH) of the guanidinium moiety of arginine residues and CH$_2$/CH$_3$ deformation modes, respectively[55,56]. This is in accordance with the fact that the structured C-terminal RBD of FUS contains 37 arginine residues that can facilitate its adsorption to the negatively charged SERS substrate, thereby resulting in a significant enhancement in our SERS spectra. Since the biomolecular condensation is context-dependent and chain–chain interactions may vary with solution conditions such as pH, we tested if single-droplet SERS was able to capture the pH-induced changes within the condensates. These measurements showed that the structural content increased upon lowering the pH from pH 8.5 to 5.5 indicating an increase in the phase separation propensity at a lower pH possibly due to increased protein-protein interactions (Supplementary Fig. 5). Such a structural transition during phase separation has also been observed for a FUS-like protein, TDP-43[57]. Taken together, this set of single-droplet SERS illuminated the inner conformational details within FUS condensates. We next asked whether this ultra-sensitive tool can be utilized to elucidate the structural details of FUS-RNA heterotypic condensates.

## Illuminating FUS-RNA heterotypic condensates using single-droplet SERS

RNA is known to modulate the phase behavior and biophysical properties of liquid condensates formed by several RNA-binding proteins including FUS[4,58,59]. With the objective to elucidate the effect of polyU RNA on the chain conformations within the droplets, we performed normal Raman and SERS at different stoichiometries of RNA and protein (Fig. 5a, b and Supplementary Fig. 6). A careful inspection of normal single-droplet Raman spectra showed two RNA marker bands, a shoulder band at 782 cm$^{-1}$ and a band at 1230 cm$^{-1}$ corresponding to uracil breathing and ring stretching modes, respectively (Fig. 5c, d)[60]. This is corroborated by the Raman difference spectrum as well (Fig. 5e). An increase in the RNA concentration leads to its greater recruitment within the phase-separated droplets which is confirmed by the linear plot of peak intensity at 1230 cm$^{-1}$ as a function of RNA concentration (Fig. 5f). Additionally, these single-droplet Raman measurements also allowed to obtain the stoichiometry of RNA and protein within

condensates by following the ratio of intensities at 1230 cm$^{-1}$ (uracil ring stretching of RNA) to at 1450 cm$^{-1}$ (CH$_2$/CH$_3$ deformation modes of protein). A linear relationship obtained can be used as a calibration line to evaluate the stoichiometry of RNA and protein within the condensed phase (Fig. 5g). Such quantitative and ratiometric estimates can be valuable in determining the concentration and composition of complex multi-component and multi-phasic condensates. Upon a closer inspection of the amide I region, we observed a considerable blue shift (Fig. 5h) from 1671 cm$^{-1}$ to 1682 cm$^{-1}$ indicating a β→disorder conversion with an increase in RNA concentration. This could potentially be due to the formation of more liquid-like condensates having more disorder and less β-content at higher RNA concentrations. Since the uracil carbonyl (C(4)=O) stretching mode can appear around this region[61], we next zoomed into the amide III region to independently confirm this unraveling of FUS in the presence of RNA resulting in a more liquid-like behavior of FUS-RNA heterotypic condensates (Fig. 5i). In this amide III region as well, in the absence of RNA, we observed primarily random coils (~1262 cm$^{-1}$) with a small contribution of β-structure (broad shoulder at ~1248 cm$^{-1}$) that disappeared at higher RNA to protein stoichiometry. This observation also supports that the observed blue shift in the amide I might be due to an increase in the disordered content rather than the C(4)=O stretching mode of the uracil ring. As in the case of single-droplet normal Raman experiments of FUS droplets in the absence of RNA, we see a variation in intensities of several Raman bands in the presence of varying RNA concentrations suggesting a conformational heterogeneity within the condensed phase (Supplementary Fig. 6). Additionally, we observed lower α-helical contents within the droplets (amide III: 1325 cm$^{-1}$; amide I: 1660 cm$^{-1}$) in the presence of RNA as shown in the Raman difference spectrum (Fig. 5e). This observation also hinted at a possible unwinding of the helical region of RBD. We next set out to characterize these conformational changes within the liquid condensates as a function of RNA concentration using ultra-sensitive SERS.

Prior to performing SERS within droplets at various concentrations of RNA, we carried UV–vis absorption spectroscopy that established the stability of nanoparticles in the presence of RNA (Supplementary Fig. 7a, b). Our turbidity and imaging assays showed that SERS substrate does not alter the behavior of FUS-RNA droplets (Supplementary Fig. 7c, d). Confocal microscopy imaging revealed complete encapsulation of nanoparticles within these droplets. We next set out to record single-droplet SERS (Fig. 6a and Supplementary Fig. 7e). Figure 6b depicts the stacked SERS spectra from individual droplets at varying concentrations of RNA. Interestingly, we observed a broad amide I band centered at 1682 cm$^{-1}$ that was undetected in SERS within FUS-only droplets. This amide I peak represents disordered polypeptide conformers with some of the β structures within the FUS-RNA droplets which is also depicted in the Raman difference plot (Fig. 6c). We believe that the interactions between the negatively charged phosphate backbone of RNA and positively charged C-terminal RBD alter the orientation of polypeptide chains on the surface of nanoparticles which brings the C=O groups of the polypeptide backbone in proximity to the nanoparticle surface for enhancement to occur. A closer inspection of the amide III region

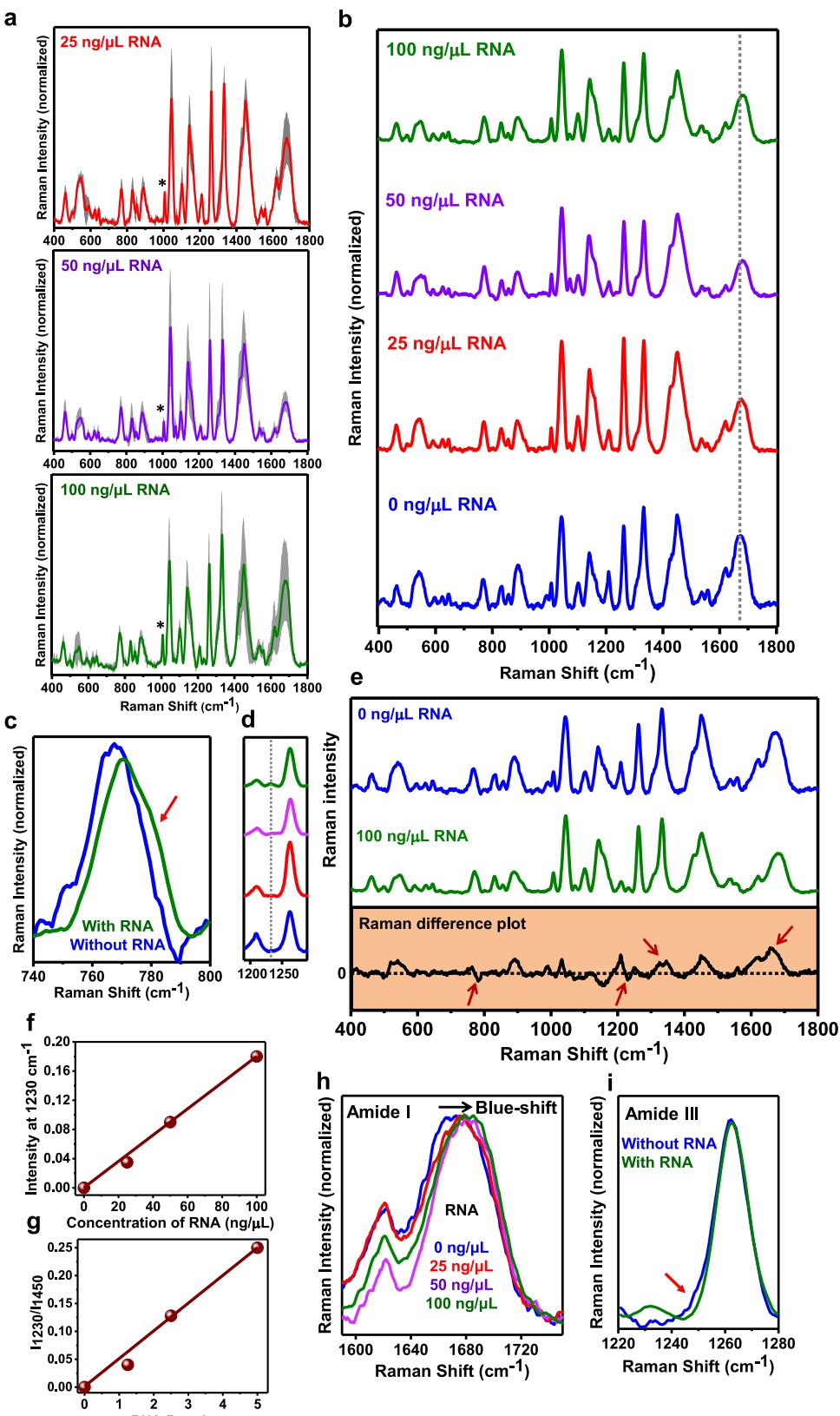

showed a broad band centered at 1245 cm$^{-1}$ corresponding to β-rich and disordered/extended conformations and a band at 1300 cm$^{-1}$ corresponding to α-helical structures for all the RNA concentrations. We observed that with an increase in RNA concentration, there is a decrease in the intensity of the amide III band at 1300 cm$^{-1}$ implying a reduction in the overall α-helical content indicating RNA-induced structural loss in FUS-RNA condensates (Fig. 6d). Interestingly, a careful inspection of the skeletal C-C stretching mode of α-helical structures at 940 cm$^{-1}$ showed a marked reduction as a function of RNA (Fig. 6e). This is probably because the electrostatic interactions between the protein and RNA disrupt the proposed cation–π interactions between tyrosine residues in the LC domain and arginine residues

**Fig. 5 | Single-droplet normal Raman spectra of FUS in the presence of RNA.**
**a** Average single-droplet normal Raman spectra in the presence of 25 ng/µL,
50 ng/µL, and 100 ng/µL poly-U RNA (spectra recorded at 500 mW laser power
with a ×100 objective; number of droplets, $n = 3$). Solid lines represent the mean,
whereas shaded region represents the standard deviation ($n = 3$). All spectra are
normalized with respect to the phenylalanine ring breathing band at 1003 cm$^{-1}$
marked by an asterisk. **b** Stacked average single-droplet normal Raman spectra
for different concentrations of RNA are shown in blue (0 ng/µL RNA), red (25 ng/
µL RNA), purple (50 ng/µL RNA), and olive (100 ng/µL RNA) for comparison (the
dotted line shows the shift in amide I in the presence of RNA). **c** Shoulder band at
-782 cm$^{-1}$ indicated by a red arrow corresponding to the uracil ring breathing
mode for FUS droplets in the presence of 100 ng/µL RNA. **d** Another RNA marker
band at -1230 cm$^{-1}$ corresponding to the uracil ring stretching mode. **e** Raman

difference plot (between 0 ng/µL RNA and 100 ng/µL RNA) of single-droplet
normal Raman spectra of droplets in the absence and presence of 100 ng/µL RNA
(arrows indicate the differences of interest). Arrows at 780 cm$^{-1}$ and 1230 cm$^{-1}$
represent the RNA marker bands. Arrows at 1325 cm$^{-1}$ and 1660 cm$^{-1}$ denotes
greater α-helical content within FUS droplets in the absence of RNA. **f** Linear plot
of RNA marker band at 1230 cm$^{-1}$ versus concentration of RNA (ng/µL) used.
**g** Plot of ratio of intensity at 1230 cm$^{-1}$ and 1450 cm$^{-1}$ versus RNA: protein.
**h** Zoomed in amide I region for FUS droplets at different concentrations of RNA
indicating blue-shift of band maxima. **i** Zoomed in amide III region band for FUS
droplets in the absence and presence of RNA (100 ng/µL). Arrow indicates a
shoulder at -1248 cm$^{-1}$ denoting small contribution of β-sheets in addition to
random-coil structures within droplets in the absence of RNA. See "Methods" for
details of data acquisition, processing, and analysis.

in the RBD[14,62]. We propose that the interaction between a polyanion
and the FUS increases the intrinsic disorder within the polypeptide
chains at the expense of α-helical structures. Moreover, we observed
changes in intensities of several vibrational modes associated with
aromatic residues, tyrosine, and tryptophan at 683, 724, 749, 800, 915,
1188, 1588, 1621 cm$^{-1}$ which indicate the changes in the orientation of
the aromatic ring of these residues on the nanoparticle surface in the
presence of RNA. Deconvolution and analysis of the band corre-
sponding to N–H deformations of the guanidinium moiety of arginine
and shoulder band for CH$_2$/CH$_3$ deformations indicated a reduction in
the enhancement of arginine residues with an increase in the RNA
concentration (Fig. 6f, Supplementary Table 2). This observation
directly captures the interaction between RNA and FUS by modulating
the polypeptide orientation on the SERS substrate surface. Taken
together, our single-droplet SERS results illuminate some key struc-
tural details within FUS-RNA condensates and highlight RNA-mediated
partial unwinding of the structured domains in the C-terminal RBD.

## Discussion

We developed an ultra-sensitive single-droplet Raman spectroscopic
methodology to elucidate the inherent conformational heterogeneity
and structural distribution within biomolecular condensates of FUS in
a droplet-by-droplet manner. This unique methodology combines the
capabilities of vibrational spectroscopy and optical microscopy
offering a wealth of molecular information within the mesoscopic
liquid condensed phase at the single-droplet resolution. Normal
vibrational Raman spectroscopy can probe the detailed molecular
structure and conformational reorganizations of the internal and
external components of individual liquid droplets[63]. However,
recording detailed vibrational signatures from liquid states is highly
challenging due to a low Raman scattering cross-section of proteins[27].
Therefore, such measurements lack adequate sensitivity and often
require unusually high concentrations, laser power, and magnifica-
tions. Such requirements can be detrimental to soft biological samples
and lead to laser-induced damage and other artifacts. These limitations
can be elegantly overcome by surface-engineered metal nanoparticle-
induced plasmonic enhancements. The electrostatic interaction
between positively charged polypeptide chains of FUS and negatively
charged iodide-coated silver nanoparticles causes significant plas-
monic enhancement of certain protein vibrational modes. Focusing a
low-power laser beam into each droplet encapsulating surface-coated
plasmonic nanostructures permitted us to record the Raman scatter-
ing bands arising due to different vibrational modes from the meso-
scopic protein-rich droplets. We took advantage of the versatility of
this technique to capture complex conformational characteristics of
heterotypic FUS-RNA condensates at a single-droplet resolution.
However, we would like to state some limitations of this methodology.
The most critical step in our single-droplet SERS experiments is the
preparation of silver plasmonic nanostructures. Since this method
involves close contact between the enhancing surface and the analyte,

removing the surface impurities is important for the homogeneity and
reproducibility of the SERS signal. Additionally, different proteins
carry different surface charges; therefore, the same SERS substrate
may not work to a similar extent for all the phase-separating systems.
Enhancements at a higher ionic strength can be inadequate due to
weaker electrostatically mediated protein-nanoparticle interactions.
Moreover, plasmonic substrates have limited re-usability and selec-
tivity for a given analyte.

Our single-droplet Raman results showed an increase in the
structural heterogeneity within liquid droplets of FUS. Several aro-
matic amino acid residues such as tyrosine and tryptophan residues
display characteristics of the conformationally restricted environment
in the condensed phase hinting at intermolecular π–π and/or cation–π
interactions within liquid phase condensates. Our methodology
allowed us to capture the unique spectral markers for droplets formed
in the presence of varying RNA-protein ratios and showed that FUS
binds stoichiometrically to RNA as shown previously[64] and this can be
used to estimate the stoichiometry of other complex biomolecular
condensates of proteins and nucleic acids. The sensitivity of the single-
droplet Raman methodology can be enhanced ≥ 10⁴-fold in the SERS
format. There is limited understanding of the protein–RNA interaction
mechanism and RNA-induced modulation of protein phase behavior[64].
Our SERS spectra showed that the C-terminal RBD undergoes a partial
unwinding in the presence of RNA together with the reduction in the
enhancement of arginine residues. This unraveling of the ordered
region in the RBD increases the polypeptide chain disorder that can
promote both homotypic (FUS-FUS) and heterotypic (FUS-RNA)
interactions within the condensed phase. Previous single-molecule
FRET results showed that highly dynamic and multivalent FUS-RNA
interactions increase the internal fluidity of the heterotypic con-
densates in the presence of RNA[64,65]. Several disease-related arginine
mutations (R216C, R244C, R514G, etc.) are known to perturb the RNA-
dependent phase behavior due to weaker protein-RNA interactions[64].
On the other hand, a key pathological glycine mutation (G156E) does
not significantly affect the phase separation propensity of FUS but can
alter material properties of condensates[65]. Future studies will aim at
addressing such important issues through the lens of vibrational
Raman spectroscopy.

In summary, single-droplet SERS allows us to zoom into the
mesoscopic condensed phase to unmask the molecular determinants
governing the intriguing condensate biophysics. This potent metho-
dology also offers a unique capability and adaptability by using dif-
ferent surface functionalities and other metals for enhancements of
unique sets of vibrational bands. Additionally, cellular uptake of these
engineered nanoparticles can open new avenues to study intracellular
phase transitions using vibrational spectroscopy. Such advancements
will pave the way for ultra-sensitive detection, characterization, and
quantification of a wide range of biomolecular condensates involved in
physiology and disease as well as in emerging applications in drug
delivery and synthetic biology.

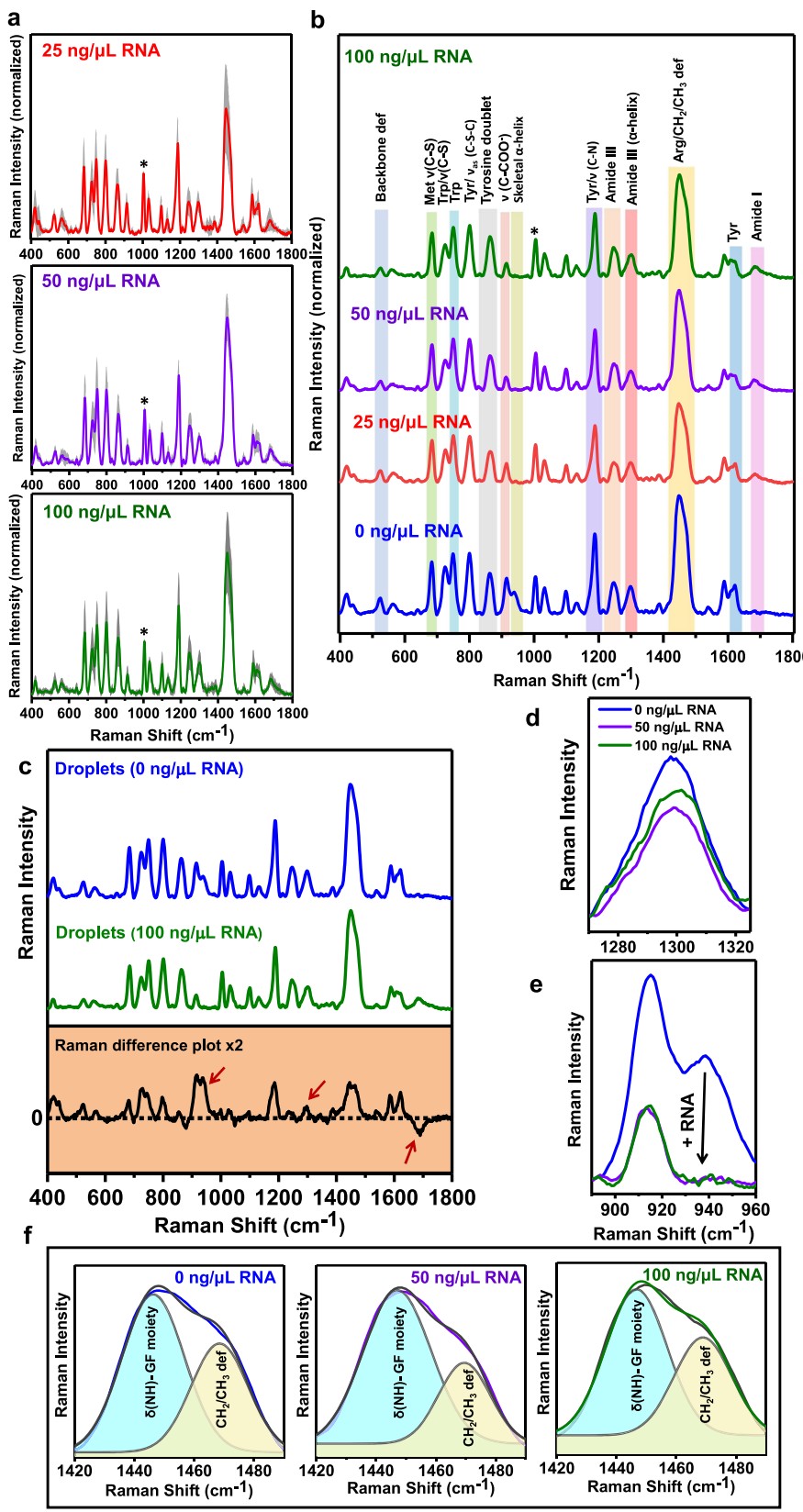

## Methods

### Recombinant protein expression, purification

The plasmid expressing MBP-*Tev*-FUS-*Tev*-His₆ was transformed into *E. coli* BL21(DE3) RIPL strain. The recombinant FUS was overexpressed and then purified using affinity chromatography[37]. For overexpression, cultures were grown in LB media at 37 °C, 220 rpm till O.D.₆₀₀ reached 0.8–1 and was induced with 0.1 mM isopropyl-β-thiogalactopyranoside (IPTG) at 12 °C for 22 h. Cell pellets were obtained by centrifugation at 4 °C, 3220 × *g* for 40 min, and stored at −80 °C for further use. For purification, pellets were resuspended in lysis buffer (50 mM sodium phosphate, 300 mM NaCl, 40 mM imidazole, 10 μM ZnCl₂, 4 mM BME, and 10% v/v glycerol, pH 8.0) and cell lysis was done using probe

**Fig. 6 | Single-droplet SERS in the presence of RNA. a** Average single-droplet SERS spectra in the presence of 25 ng/μL, 50 ng/μL, and 100 ng/μL polyU RNA (spectra recorded at 5 mW laser power with a ×50 objective; number of droplets, n = 7). Solid lines represent mean, whereas shaded region represents the standard deviation (n = 7). All spectra are normalized with respect to the phenylalanine ring breathing band at 1003 cm⁻¹ marked by an asterisk. **b** Stacked average single-droplet SERS spectra for different concentrations of RNA are shown in blue (0 ng/μL RNA), red (25 ng/μL RNA), purple (50 ng/μL RNA), and olive (100 ng/μL RNA) for comparison. **c** Raman difference plot (between 0 ng/μL RNA and 100 ng/μL RNA) of single-droplet SERS spectra of droplets in the absence and presence of RNA (100 ng/μL). An arrow at 1682 cm⁻¹ represents the emergence of amide I at

higher RNA concentrations and arrows at 940 cm⁻¹ and 1300 cm⁻¹ represents greater α-helical content within droplets in the absence of RNA. **d** Zoomed in Amide III region for FUS droplets in the absence and presence of RNA (50 ng/μL, 100 ng/μL). **e** Skeletal C–C stretching mode of α-helical structures at 940 cm⁻¹ that disappears at higher RNA concentrations. **f** Gaussian deconvolution of the region 1420–1490 cm⁻¹. The black line represents the actual data while the colored lines represent the cumulative fit. Cyan region represents the N–H deformations of the guanidinium fragment (GF) of arginine residues, while the light-yellow region represents the CH₂/CH₃ deformations. See Supplementary Table 2 for percentage analysis. See "Methods" for details of data acquisition, processing, and analysis.

sonication at 5% amplitude, 15 s ON and 10 s OFF for 25 min. The lysate was centrifuged at 4 °C, 15,557 × g for 1 h, followed by incubation of the supernatant with equilibrated Ni-NTA agarose beads for 1.5 h at 4 °C. The beads were washed and protein was eluted with 250 mM imidazole followed by binding to the amylose column. Protein was eluted with 20 mM maltose elution buffer (50 mM sodium phosphate, 800 mM NaCl, 40 mM imidazole, 10 μM ZnCl₂, 20 mM maltose and 1 mM 1,4-dithiothreitol, pH 8.0). The concentration of the protein was estimated by measuring absorbance at 280 nm ($\varepsilon_{280\ nm}$ = 1,30,670 M⁻¹ cm⁻¹) and samples were run on an SDS-PAGE gel to confirm the purity of the protein. The purified protein was further stored at 4 °C for future use.

The plasmid containing His₆-tagged TEV protease was transformed into *E. coli* strain BL21(DE3) plysS. Cells were grown at 37 °C, 220 rpm, and overexpression was induced by 0.35 mM IPTG at 16 °C for 20 h. Cultures were pelleted and stored at −80 °C for further use. The pellets were thawed at 30 °C and suspended in lysis buffer (25 mM HEPES, 150 mM NaCl, 20 mM KCl, and 20 mM MgCl₂, pH 7.4) along with phenylmethylsulfonyl fluoride and lysozyme to enhance cell lysis which was carried out by probe sonication (5% amplitude, 15 s ON/10 s OFF for 25 min). The soluble fraction was separated by centrifugation and the supernatant was passed twice through a pre-equilibrated Ni-NTA column at 4 °C. The beads were washed with wash buffer (lysis buffer + 20 mM imidazole) and protein was eluted with 300 mM imidazole and dialyzed against buffer without imidazole, overnight at 4 °C. Protein was concentrated using a 10 kDa MWCO filter and stored at −80 °C for further use.

**Fluorescence labeling**

For labeling, purified FUS was concentrated using a 50 kDa MWCO amicon filter, and incubated with 0.3 mM tris(2-carboxyethyl) phosphine (TCEP) for 30 min on ice following which reactions with 1:4.5 molar ratio of protein: dye (for AlexaFluor488-C5-maleimide) and 1:30 (for Fluorescein-5-maleimide) were set up in native buffer at 25 °C and kept under shaking for 2 h in dark. Unreacted dye was removed by buffer exchange using 50 kDa MWCO amicon filters. Labeling efficiency was calculated by measuring absorbance at 280 nm ($\varepsilon_{280\ nm}$ = 1,30,670 M⁻¹cm⁻¹, for full-length FUS) and 494 nm ($\varepsilon_{494}$ = 72,000 M⁻¹ cm⁻¹, for AlexaFluor488-C5-maleimide and $\varepsilon_{494}$ = 68,000 M⁻¹ cm⁻¹, for F-5-M) to estimate total protein concentration and labeled protein concentration.

**Phase separation assays**

Phase separation of FUS was initiated by TEV cleavage in a 1:10 molar ratio (TEV:protein) at room temperature in 20 mM sodium phosphate, pH 7.4. Turbidity of phase-separated samples was then estimated using 96-well NUNC optical bottom plates (Thermo Scientific) on a Multiskan Go (Thermo Scientific) plate reader by recording the absorbance at 350 nm. The protein concentration for all the experiments was fixed to 20 μM along with 0.1 nM of iodide-modified silver nanoparticles Ag IMNPs (for reactions in the presence of nanoparticles). For phase separation in the presence of RNA, LLPS was induced in the presence of 25 ng/μL, 50 ng/μL, and 100 ng/μL polyU RNA with or without 0.1 nM

Ag IMNPs. Single-stranded polyU RNA was obtained from Sigma-Aldrich with a molecular weight of 800–1000 kDa. RNA concentration was estimated using a Genova Life Science spectrophotometer (ver.1.51.4). The total sample volume used was 100 μL for all the measurements and then background subtracted turbidity was plotted using Origin.

**Confocal microscopy**

Confocal fluorescence imaging of FUS droplets with and without Ag IMNPs was performed on ZEISS LSM 980 Elyra 7 super-resolution microscope using a ×63 oil-immersion objective (Numerical aperture 1.4). For visualizing droplets of FUS, 200 nM (1%) of Alexa488 or F-5-M labeled protein was doped with unlabeled protein, and 2–3 μL of the freshly phase-separated sample was placed into a chamber made on a glass slide (Fisher Scientific 3″ × 1″ × 1 mm). The chamber made by using double-sided tape was then sealed with a square coverslip to avoid evaporation of the sample. For visualization of encapsulated Ag IMNPs (0.1 nM), Alexa488-labeled protein was imaged using a 488-nm laser diode (11.9 mW), and Ag IMNPs were imaged using a 405-nm laser diode (11.9 mW). For images captured through the eyepiece, a metal halide lamp was used to excite both labeled protein and nanoparticles. All the confocal images were then processed and analyzed using ImageJ (NIH, Bethesda, USA).

**Fluorescence recovery after photobleaching (FRAP) measurements**

FRAP experiments for droplets with and without Ag IMNPs were performed on ZEISS LSM 980 Elyra 7 super-resolution microscope using a ×63 oil-immersion objective (Numerical aperture 1.4). All the FRAP experiments were performed using 200 nM (1%) of Alexa488-labeled protein. The recovery of the chosen region of interest (ROI) after photobleaching using a 488-nm laser was then recorded using ZEN Pro 2011 (ZEISS) software provided with the instrument. The fluorescence recovery curves were then normalized and plotted after background correction using Origin.

**Sedimentation assays**

The absence of MBP within FUS droplets was confirmed using sedimentation assay. The MBP-FUS was cleaved by TEV protease to induce phase separation and after 20 min the reaction was centrifuged at 25,000 × g, 25 °C for 30 min to pellet down all the droplets (condensed phase). Both supernatant and the pellet were then separated carefully and the pellet was dissolved in 8 M urea. Samples were then heated and run on 12% SDS-PAGE along with the respective controls.

**Circular dichroism (CD) measurements**

CD measurements were performed on a BioLogic scanning MOS-500 CD spectrometer using a quartz cuvette of 1 mm path length. The final protein concentration of 5 μM protein and 0.025 nM Ag IMNPs in 20 mM sodium phosphate buffer pH 7.4 was used for recording the data. The spectra were averaged over three scans and blank subtracted data were plotted using Origin software.

## Size distribution analysis of droplets

Droplet reactions of 20 µM FUS (with 1% Alexa488-labeled FUS) were set up in the absence and the presence of 0.1 nM Ag IMNPs. Confocal imaging was performed at same time points for both the samples using ZEISS LSM 980 Elyra 7 super-resolution microscope with a ×63 oil-immersion objective (Numerical aperture 1.4). For visualization of the fluorescently-labeled droplets, a 488-nm laser diode (11.9 mW) and for the Ag IMNPs, a 405-nm laser diode (11.9 mW) was used. Images were obtained from multiple independent reactions and were analyzed using ImageJ (NIH, Bethesda, USA). A distribution of projection area of droplets was constructed and plotted as a distribution frequency plot. The mean droplet diameter was estimated from the mean area.

## Preparation of silver nanoparticles

Silver nanoparticles were prepared by the Lee-Meisel method[48]. Initially, 8.49 mg of silver nitrate was dissolved in 50 mL of filtered milli-Q water and stirred vigorously (1000 rpm) at its boiling point (-98 °C) for 30 min. One and five-tenths milliliters of freshly prepared 1% (w/v) aqueous trisodium citrate was added to the reaction mixture dropwise and further stirred for additional 30 min till the color changed to yellow–green. The solution was cooled down to room temperature and was further characterized using UV–visible absorption spectroscopy, transmission electron microscopy, and zeta (ζ) potential measurements.

## Preparation of iodide-modified silver nanoparticles

One milliliter of silver nanoparticles was centrifuged in a 1.5 mL microcentrifuge tube at 2320 × g for 15 min at room temperature. The supernatant was discarded and the resulting colloid was resuspended in 1 mL of Milli-Q water and centrifuged again. The resulting colloidal suspension (50 µL) was then mixed with an equal volume of 12 mM potassium iodide (KI) and incubated for 24 h at room temperature in dark. After incubation, the resulting iodide-modified nanoparticles were centrifuged at 2320 × g for 10 min at room temperature and resuspended in 100 µL of milli-Q water. Resulting Ag IMNPs were characterized using UV–visible absorption spectroscopy, transmission electron microscopy, and zeta (ζ) potential measurements.

## UV–visible absorption spectroscopy

All the UV–vis absorption spectra were collected on a Multiskan Go (Thermo Scientific) plate reader using 96-well NUNC optical bottom plates (Thermo Scientific). The total sample volume used was 100 µL for all the measurements. Twenty micromolar full-length FUS was used whereas the concentration of Ag IMNPs used was fixed to 0.1 nM and 25 ng/µL, 50 ng/µL, and 100 ng/µL of poly-U RNA were used. Background subtracted absorption spectra from 300–800 nm were normalized and plotted using Origin.

## Dynamic light scattering (DLS)

Zeta potential measurements for iodide-modified nanoparticles were carried out on a Malvern Zetasizer Nano ZS90 instrument (Malvern, UK) using a He-Ne laser (632 nm) as an excitation source. All the measurements were carried out at room temperature and 0.05 nM of Ag IMNPs in filtered milli-Q water was used for estimating zeta potential.

## Transmission electron microscopy (TEM)

TEM images were obtained on a Jeol JEM-F200. Three microliters of half-diluted colloidal suspension were adsorbed on a 300-mesh carbon-coated electron microscopy grid and allowed to dry overnight. Histogram for nanoparticle size distribution was created using ImageJ (NIH, Bethesda, USA) software and plotted using Origin.

## Normal Raman and single-particle surface-enhanced Raman spectroscopy (SERS)

Single-droplet normal Raman and SERS spectra were recorded on an inVia laser Raman microscope (Renishaw, UK) at -25 °C. For single-droplet normal Raman measurements droplet reaction (2 µL) of full-length FUS (20 µM) with or without RNA was placed on a glass slide covered with an aluminum foil and single droplets were focused using a ×100 long working distance objective lens (Nikon, Japan). An NIR laser (785 nm) with an exposure time of 10 sec and 500 mW (100%) laser power was used to excite the samples. Raman scattered light was collected and dispersed using a diffraction grating (1200 lines/mm) and was further detected by an air-cooled CCD detector whereas the Rayleigh scattered light was blocked using an edge filter of 785 nm. For single-droplet SERS measurements, phase separation of full-length FUS (20 µM) was set up in the presence of 0.1 nM Ag IMNPs with or without RNA, and 2 µL of the droplet reaction was placed on a glass slide covered with an aluminum foil. Single droplets were focused using a ×50 long-working-distance objective lens (Nikon, Japan) and an NIR laser (785 nm) with an exposure time of 10 s and 5 mW (1%) laser power was used to excite the samples. Experiments were repeated with different batches of freshly purified protein and freshly prepared nanoparticles. Data was acquired using Wire 3.4 software provided with the Raman spectrometer. The collected Raman spectra were baseline corrected using cubic spline interpolation method and smoothened using Wire 3.4 software and plotted using Origin.

## Reporting summary

Further information on research design is available in the Nature Research Reporting Summary linked to this article.

## Data availability

The data are available within the Article, Supplementary Information, and Source Data file. Source data are provided with this paper.

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

## Acknowledgements
The authors thank IISER Mohali, Department of Science and Technology (Nano-Mission grant # SR/NM/NS–1475/2014 to S.M. and FIST grant # SR/FST/LS-II/2017/97 to the Department of Biological Sciences, IISER Mohali), Science and Engineering Research Board (SUPRA grant # SPR/2020/000333 to S.M.), Ministry of Education, Govt. of India (Centre of Excellence grant # MHRD–14-0064 to S.M.) for financial support, Prof. Dorothee Dormann (Ludwig-Maximilians University and Institute für Molekulare Biology, Mainz) for her kind gift of MBP-*Tev*-FUS-*Tev*-His₆ plasmid, the TEM facility at IISER Mohali for electron microscopy, and Dr. Mily Bhattacharya (Thapar Institute) and the members of the Mukho-padhyay lab for critically reading this manuscript.

## Author contributions
S.M. conceived the project. A.A., A.J., and S.M. further developed the concept and the experimental design. A.A., A.J. A.W., and S.G.P. performed the experiments and analyses. A.A. and A.J. prepared the figures and wrote the first draft. S.M. supervised the work, edited the manuscript, obtained funding, and provided the overall direction. All authors discussed the results and commented on the manuscript.

## Competing interests
The authors declare no competing interests.
