## [Peer Review File · Nature Communications]

REVIEWER COMMENTS

Reviewer #1 (Remarks to the Author):

This manuscript studies the possibility of using droplets for the determination of fused sarcoma aggregates by using nanoparticles and SERS combined with fluorescence. After carefully reading the manuscript, I can say that the science involved is correct. However, and unfortunately, to me this manuscript is more incremental than new. Direct characterization of biomarker using liquid samples in many forms including spontaneous or induced formed droplets had been developed for the last 20 years. Direct detection and classification of proteins and nucleic acids, the same. Thus, in my opinion the manuscript is a collage of those technologies that for sure should be publish, but not in Nat. Comm.

Reviewer #2 (Remarks to the Author):

This is an impressive work, which addresses the challenging problem of the internal organisation of biomolecular condensates. The proposed approach is potentially powerful to elucidate the driving forces and stabilising interactions.

Major points:

- By nature, the condensates are conformationally heterogeneous. How the spectrum reflects such heterogeneity? The readout will inform only on stable contacts or also on the temporal or spatial variations of such contacts?

- What are the major limitations of the technique?

- Also by nature, the condensates are context-dependent, i.e. the interactions should vary with cellular conditions. Can the authors demonstrate changes in the spectrum upon a relevant change in the buffer composition (maybe pH or ion concentration).

- Can the authors provide some kind of easy, user-friendly guideline for the readout (for non-Raman experts)? Maybe illustrated by a figure? Or perhaps a table on the possible interactions, chemical groups, which are possible to observe?

- Condensates age with time and form aggregates. Familial mutations may accelerate this process. It would be highly exciting to show how the interactions differ between normal FUS and a mutant (eg. well-studied G156E) .

- I would be keen to see a comparison with the work of Sua Myong (Niaki et al 2020 Mol Cell, Rhine et al 2020 Mol Cell), who did FRET on the dynamics of RNA FUS interactions , also ALS_ associated mutants.

Minor comments

- The language of the paper is slightly rhetorical. It is a solid experimental work, which does not need it. For example 'molecular language' of condensates - I see this in many papers. Instead, you study interactions, or stabilising factors.

Can you simplify these expressions text (also for disease etc)? The paper would benefit from it and reflect that it is a highly solid work.

Reviewer #3 (Remarks to the Author):

Major revision requested.

Authors have probed structural features of liquid-liquid phase-separated condensates formed by Fused in Sarcoma with Raman microscopy and SERS, in the absence and the presence of the varied amount of RNA.

As rightly pointed out by the authors, there is a need of developing methods that have the spatial resolution of biological-imaging techniques but can simultaneously provide structural information about the condensates. Raman microscopy can provide a solution to this but the inherently feeble Raman cross-section of typical vibrational modes requires the use of high laser powers and consequently risks photo damages. The authors proposed to provide a solution that uses negatively charged silver nanoparticles that by getting spontaneously encapsulated within the condensed phase can generate SERS signals at much improved S/N.

The paper is well-written; however, the authors need to answer the following concerns before this can be considered for publication.

1) In figure 3, we observe that there is considerable variation in the relative intensities of the conventional Raman signatures at, $\sim 890 \text{ cm}^{-1}$ (assigned as Trp N-H bend) $\sim 1042 \text{ cm}^{-1}$ (assigned as Phe [$\delta(\text{R}(\text{CH}))$]), $\sim 1262 \text{ cm}^{-1}$ (assigned as Amide III nonregular/turns) (compare to the 1007 cm^{-1} reference mode) amongst the droplets. Please provide accounts for these variations (e.g., standard deviation) and explain. The extent of the variations may provide additional information about the inherent heterogeneity. Similarly, in supplementary figure 4, we can observe considerable variation in the peaks $\sim 830 \text{ cm}^{-1}$, 1042 cm^{-1} , and 1101 cm^{-1} , in the presence of RNA, which also need a detailed explanation.

2) Authors have probed the effect of colloidal stability of the negatively charged silver nanoparticles by performing absorption spectroscopy. In supplementary figure 3, we observe a considerable widening of the normalized absorption signature of the nanoparticles in the presence of FUS-droplets, which may be resulting from aggregation. Authors should provide a comparison of true absorption spectral intensities alongside the normalized spectra to discuss the extent of aggregation and particle loss.

3) To test the LLPS behavior in the presence of NPs, turbidity and microscopy experiments were performed. The author should provide a comparison of the distribution of droplet sizes, as obtained from microscopic studies, before and after the addition of the NPS.

4) In figure 5h, we observe that the relative intensity at $\sim 1621 \text{ cm}^{-1}$, varied at different RNA concentrations. This intensity change, however, is not linear to the added RNA concentration.

Authors have assigned this peak to the Tyr. Raman Vibrational signature in this region may also originate from β -sheet conformation. Is this pointing towards additional structural intermediates that can form at certain RNA concentration(s)?

5) Histidine sidechain has been known to form a tautomeric doublet Raman signature in the same region where authors assigned the SERS peaks to amide III. The authors need to justify this assignment, especially when the RBD contains three histidine residues.

6) Authors have attributed the decrease in the intensity of the amide III bands at 1300 cm^{-1} (In figure 6) to the RNA-induced reduction in alpha-helical content in the FUS-RNA condensates. On the other hand, the observation of the appearance of the SERS peak at 1682 cm^{-1} (only in the presence of RNA) has been linked to a plausible orientational change of the polypeptide backbone on the NP surface due to protein-RNA interaction. The author should justify their claim by discussing the expected effect of such a reorientation on the 1300 cm^{-1} peak.

7) The overall diffusional behavior within the droplet seems to be unaffected in the presence of NPs. The high negative charge on the NPs, however, may affect protein structure and protein-RNA interaction, locally.

8) Authors are, therefore, required to find the effect of NP-charge in protein-RNA interaction, on protein structure, and in determining the orientation of the protein on the NP-surface, computationally and/or by performing complementary experiments.

Point-by-point response to reviewers' comments (NCOMMS-22-05462)

Peer Reviewer #1

Reviewer Comments: This manuscript studies the possibility of using droplets for the determination of fused sarcoma aggregates by using nanoparticles and SERS combined with fluorescence. After carefully reading the manuscript, I can say that the science involved is correct. However, and unfortunately, to me this manuscript is more incremental than new. Direct characterization of biomarker using liquid samples in many forms including spontaneous or induced formed droplets had been developed for the last 20 years. Direct detection and classification of proteins and nucleic acids, the same. Thus, in my opinion the manuscript is a collage of those technologies that for sure should be publish, but not in Nat. Comm.

Authors' Response: We respectfully disagree with the comments made by this reviewer. Using a novel experimental design and single-droplet surface-enhanced Raman scattering, we have been able to unmask the key molecular determinants of biological phase separation, a fascinating research area that has emerged only recently.

Peer Reviewer #2

Reviewer Comments: This is an impressive work, which addresses the challenging problem of the internal organisation of biomolecular condensates. The proposed approach is potentially powerful to elucidate the driving forces and stabilising interactions.

Authors' Response: We thank the reviewer for her/his kind words, appreciation, insightful comments, and valuable suggestions on our work. Our responses are as follows. The changes in the revised manuscript are marked in blue.

Reviewer Comments: By nature, the condensates are conformationally heterogeneous. How the spectrum reflects such heterogeneity? The readout will inform only on stable contacts or also on the temporal or spatial variations of such contacts?

Authors' Response: We thank the reviewer for asking this question. Condensates are conformationally heterogeneous in nature, and our steady-state Raman spectra capture this heterogeneity. The Raman marker bands for the secondary structure of proteins such as the Amide I and Amide III display broadening within the condensed phase highlighting the heterogeneous conformations of polypeptide chains within liquid-like droplets. Additionally, there is a considerable variation in the relative intensities of the Raman bands amongst the droplets in the absence and presence of RNA suggesting variable contacts formed within different droplets providing additional information about the inherent heterogeneity within these condensates. Future time-resolved Raman measurements can potentially capture the temporal variations in conformational distributions.

Reviewer Comments: What are the major limitations of the technique?

Authors' Response: We thank the reviewer for asking this important question. On page 20 of our revised manuscript, we have now clearly stated the limitations of this technique that involve the surface charge, surface impurity in plasmonic nanostructures, higher ionic strength, and limited re-usability.

Reviewer Comments: Also by nature, the condensates are context-dependent, i.e. the interactions should vary with cellular conditions. Can the authors demonstrate changes in the spectrum upon a relevant change in the buffer composition (maybe pH or ion concentration).

Authors' Response: This is a very interesting point. We agree with the reviewer and have now performed single-droplet SERS experiments with varying pH. We observed that upon lowering the pH to 5.5, the phase separation propensity increases which is concomitant with the increase in the structural content of the C-terminal RBD of FUS and more protein-protein interactions at lower pH. We have now included the SERS data for different pH conditions in our revised manuscript (page 13) and Supplementary Fig. 5. Our observation is in accordance with a recent study on a FUS-like protein, TDP-43, that exhibits higher structural propensity upon phase separation (Conicella et al. *Proc. Natl. Acad. Sci. U. S. A.* 2020, 117, 5883-5894).

Reviewer Comments: Can the authors provide some kind of easy, user-friendly guideline for the readout (for non-Raman experts)? Maybe illustrated by a figure? Or perhaps a table on the possible interactions, chemical groups, which are possible to observe?

Authors' Response: This is a great suggestion and we agree that a user-friendly guideline for common Raman bands would be extremely useful for non-Raman experts. We have now included a table in Supplementary Information (Supplementary Table 1) summarizing the common vibrational bands.

Reviewer Comments: Condensates age with time and form aggregates. Familial mutations may accelerate this process. It would be highly exciting to show how the interactions differ between normal FUS and a mutant (eg. well-studied G156E).

Authors' Response: This is an interesting point. We are indeed undertaking a detailed study of the influence of several ALS/FTD-associated mutations (including G156E) by utilizing single-droplet Raman and other single-droplet spectroscopic tools we are developing in our lab. These results will be reported in a separate study.

Reviewer Comments: I would be keen to see a comparison with the work of Sua Myong (Niaki et al 2020 Mol Cell, Rhine et al 2020 Mol Cell), who did FRET on the dynamics of RNA FUS interactions, also ALS-associated mutants.

Authors' Response: This is a very pertinent point and we have now discussed previous results in the context of our findings in the Discussion section on page 20 and have also included these references (Ref. 64 & 65).

Reviewer Comments: The language of the paper is slightly rhetorical. It is a solid experimental work, which does not need it. For example, 'molecular language' of condensates- I see this in many papers. Instead, you study interactions, or stabilizing factors. Can you simplify these expressions text (also for disease etc)? The paper would benefit from it and reflect that it is a highly solid work.

Authors' Response: We agree with our reviewer. We have now replaced "molecular language" with "molecule determinants" in the title and with "fundamental molecular interactions" on page 3 in the Introduction.

We are extremely grateful to this reviewer for her/his valuable comments and suggestions that helped us improve our manuscript.

Peer Reviewer #3

Reviewer Comments: Authors have probed structural features of liquid-liquid phase separated condensates formed by Fused in Sarcoma with Raman microscopy and SERS, in the absence and in the presence of varied amount of RNA. As rightly pointed out by the authors, there is a need of developing methods that have the spatial-resolution of biological-imaging techniques but can simultaneously provide structural information about the condensates. Raman microscopy can provide a solution to this but the inherently feeble Raman cross-section of typical vibrational modes, require the use of high laser powers and consequently risks photo-damages. Authors proposed to provide a solution that uses negatively charged silver nanoparticles that by getting spontaneously encapsulated within the condensed phase can generate SERS-signals at much improved S/N. The paper is well-written; however, the authors need to answer the following concerns before this can be considered for publication.

Authors' Response: We thank the reviewer for her/his kind words, appreciation, and valuable suggestions on our work. Our responses are as follows. The changes in the revised manuscript are marked in blue.

Reviewer Comments: In figure 3, we observe that there is considerable variation in the relative intensities of the conventional Raman signatures at, ~ 890 cm⁻¹ (assigned as Trp N-H bend) ~1042 cm⁻¹ (assigned as Phe [$\delta(R(CH))$]), ~1262 cm⁻¹ (assigned as Amide III nonregular/turns) (compare to the 1007 cm⁻¹ reference mode) amongst the droplets. Please provide accounts for these variations (e.g., standard deviation) and provide explanation. The extent of the variations may provide additional information about the inherent heterogeneity. Similarly, in supplementary figure 4, we can observe considerable variation in the peaks ~830 cm⁻¹, 1042 cm⁻¹, 1101 cm⁻¹, in the presence of RNA, which also need detailed explanation.

Authors' Response: We agree with the reviewer and thank her/him for raising this point. We have now included the standard deviation for these spectra (and Supplementary Fig. 2 and 6). We have also included the explanation of variability from droplet to droplet arising due to varied protein-protein interactions, conformational heterogeneity, and structural distributions (pages 8-9 and 16).

Reviewer Comments: Authors have probed the effect of colloidal stability of the negatively charged silver nanoparticles by performing absorption spectroscopy. In the supplementary figure 3, we observe considerable widening of normalized absorption signature of the nanoparticles in the presence of FUS-droplets, which may be resulting from aggregation. Authors should provide comparison of true absorption spectral intensities alongside the normalized spectra to discuss the extent of aggregation and particle loss.

Authors' Response: This is a very important point. The absorption spectra indicate nanoparticle aggregation that creates the hotspots required for enhancement. We have now included the unnormalized absorption spectra along with the normalized plots (Supplementary Fig. 4a,b) and we have explained it more clearly on page 10 of our revised manuscript.

Reviewer Comments: To test the LLPS behaviour in the presence of NPs, the turbidity and microscopy experiments were performed. The author should provide a comparison of the distribution of droplet sizes, as obtained from the microscopic studies, before and after the addition of the NPs.

Authors' Response: This is a very nice suggestion. We observe that there is little or no difference in the size of the droplets before and after the addition of NPs. We have now included the size distribution

analysis and representative microscopic images in the revised manuscript (Supplementary Fig. 4h) and explained it clearly on page 10 as well as in the Methods section (page 24).

Reviewer Comments: In the figure 5h, we observe that the relative intensity at $\sim 1621\text{ cm}^{-1}$, varied greatly at different RNA concentration. This intensity change, however, is not linear to the added RNA concentration. Authors have assigned this peak to the Tyr. Raman Vibrational signature at this region may also originate from β -sheet conformation. Is this pointing towards additional structural intermediates that can form at certain RNA concentration(s)?

Authors' Response: We thank the reviewer for asking this question. Although the peak at 1621 cm^{-1} can arise due to H-bonded anti-parallel β -sheets, previous single-molecule FRET experiments (Ref. 64 & 65) and our current Raman investigation (Amide I) revealed that in the presence of RNA, FUS forms more liquid-like condensates containing a high degree of structural disorder without having any signature of β -rich conformation. Additionally, FUS contains 36 tyrosine residues, and therefore, we believe that the peak at 1621 cm^{-1} is likely to be from the ring stretching mode of tyrosine residues.

Reviewer Comments: Histidine sidechain has been known to form a tautomeric doublet Raman signature in the same region where authors assigned the SERS peaks to amide III. The authors need to provide justification for this assignment, especially when the RBD contains three histidine residues.

Authors' Response: We thank the reviewer for raising this point. Histidine is known to form three pairs of N_{τ} -H and N_{π} -H tautomeric doublets at $983/1004$, $1282/1260$, and $1568/1585\text{ cm}^{-1}$ (N_{τ} and N_{π} are the two nitrogen atoms in the five-membered imidazole ring). In our single-droplet SERS experiments, we observe backbone Amide III at 1246 cm^{-1} and 1298 cm^{-1} which we assign to β -sheets and α -helical structures, respectively. Additionally, we would like to state that there are only 3 Histidine residues in FUS which has 526 residues (*i.e.*, 525 amide bonds); therefore, the contribution from Histidine, if at all, will be much smaller compared to Amide III. We have now included this in our revised manuscript (page 13). We are grateful to the reviewer for raising this point.

Reviewer Comments: Authors have attributed the decrease in the intensity of the amide III band at 1300 cm^{-1} (In figure 6) to the RNA-induced reduction in alpha-helical content in the FUS-RNA condensates. On the other hand, the observation of appearance of the SERS peak at 1682 cm^{-1} (only in the presence of RNA) has been linked to a plausible orientational change of the polypeptide backbone on the NP surface due to protein-RNA interaction. The author should justify their claim by discussing the expected effect of such a reorientation on the 1300 cm^{-1} peak.

Authors' Response: We thank the reviewer for pointing this out. We would like to state here that we do not see Amide I in our single-droplet SERS experiments possibly due to the interaction of the bulky side-chains of FUS with the nanoparticles that pushes the C=O group away from the critical near-field required for the plasmonic enhancement. Amide I band starts to appear with increasing RNA concentrations which we attribute to disordered polypeptide conformations. We would like to note that our single-droplet normal Raman experiments (without nanoparticles) in the presence of RNA indicate that α -helicity decreases with an increase in the RNA concentration that is shown in the Raman difference plot (Amide I and Amide III regions in Figure 5e). Taken together, a reduction in the intensity of Amide III at 1300 cm^{-1} in our SERS experiments can be attributed to the structural unwinding of the polypeptide chain in the presence of RNA. We, however, cannot rule out minor contributions from the orientational changes.

Reviewer Comments: The overall diffusional behaviour within the droplet seems to be unaffected in the presence of NPs. The high negative charge on the NPs, however, may affect protein structure and protein-RNA interaction, locally. Authors are, therefore, required to find the effect of NP-charge in protein-RNA interaction, on protein structure, and in determining the orientation of the protein on the NP-surface, computationally and/or by performing complementary experiments.

Authors' Response: This is a very nice suggestion. Performing simulations on such a large unstructured protein in the presence of nanoparticles is computationally very challenging and is beyond scope of our expertise. We have now performed additional experiments to support this. Our new CD experiments on FUS in the presence and absence of nanoparticles showed that nanoparticles indeed do not significantly alter the overall protein secondary structure. We have now included these CD spectra in our revised manuscript (Page 10 and Supplementary Fig. 4d). Additionally, our turbidity plot shown in Supplementary Figure 7c shows that the turbidity of FUS-RNA is almost similar in the absence and presence of nanoparticles indicating that the nanoparticles do not significantly alter the protein-RNA interactions and the overall phase behavior.

We are extremely grateful to this reviewer for her/his valuable comments and suggestions that helped us improve our manuscript.

REVIEWERS' COMMENTS

Reviewer #2 (Remarks to the Author):

The SERS method proposed in the paper will importantly contribute to the field of liquid-liquid phase separation of proteins. In particular to our understanding of how conformational properties relate to physiological and pathological behaviors of protein droplets.

In the revised version, the authors have addressed all my points. In this format, I recommend publication of the manuscript.